# Physical Activity Levels and Related Energy Expenditure during COVID-19 Quarantine among the Sicilian Active Population: A Cross-Sectional Online Survey Study

**Valerio Giustino** [1], **Anna Maria Parroco** [1], **Antonio Gennaro** [2], **Giuseppe Musumeci** [3,4,5], **Antonio Palma** [1,2] and **Giuseppe Battaglia** [1,2,*]

1   Department of Psychology, Educational Science and Human Movement, University of Palermo, 90144 Palermo, Italy; valerio.giustino@unipa.it (V.G.); annamaria.parroco@unipa.it (A.M.P.); antonio.palma@unipa.it (A.P.)
2   Regional Sports School of CONI Sicilia, 90141 Palermo, Italy; prof.gennaro@virgilio.it
3   Department of Biomedical and Biotechnological Sciences, Anatomy, Histology and Movement Sciences Section, School of Medicine, University of Catania, 95123 Catania, Italy; g.musumeci@unict.it
4   Research Center on Motor Activities (CRAM), University of Catania, 95123 Catania, Italy
5   Department of Biology, Sbarro Institute for Cancer Research and Molecular Medicine, College of Science and Technology, Temple University, Philadelphia, PA 19122, USA
*   Correspondence: giuseppe.battaglia@unipa.it

**Abstract:** Background: During the coronavirus disease 2019 (COVID-19) pandemic, the Italian government has adopted containment measures to control the virus's spread, including limitations to the practice of physical activity (PA). The aim of this study was to estimate the levels of PA, expressed as energy expenditure (MET–minute/week), among the physically active Sicilian population before and during the last seven days of the COVID-19 quarantine. Furthermore, the relation between this parameter and specific demographic and anthropometric variables was analyzed. Methods: 802 Sicilian physically active participants (mean age: 32.27 ± 12.81 years; BMI: 23.44 ± 3.33 kg/m$^2$) were included in the study and grouped based on gender, age and BMI. An adapted version of the International Physical Activity Questionnaire—short form (IPAQ-SF) was administered to the participants through an online survey. The Wilcoxon signed-rank test and the Kruskal-Wallis rank-sum test were used for statistical analyses. Results: As expected, we observed a significant decrease of the total weekly energy expenditure during the COVID-19 quarantine ($p < 0.001$). A significant variation in the MET–min/wk in the before quarantine condition ($p = 0.046$) and in the difference between before and during quarantine ($p = 0.009$) was found for males and females. The male group decreased the PA level more than the female one. Moreover, a significant difference in the MET–min/wk was found among groups distributions of BMI ($p < 0.001$, during quarantine) and of age ($p < 0.001$, both before and during quarantine). In particular, the highest and the lowest levels of PA were reported by the young and the elderly, respectively, both before and during quarantine. Finally, the overweight group showed the lowest level of PA during quarantine. Conclusion: Based on our outcomes, we can determine that the current quarantine has negatively affected the practice of PA, with greater impacts among males and overweight subjects. In regards to different age groups, the young, young adults and adults were more affected than senior adults and the elderly.

**Keywords:** COVID-19; coronavirus; 2019-nCoV; pandemic; quarantine; lockdown; physical activity; physical inactivity; exercise; training; home exercise; home training

## 1. Introduction

The coronavirus disease 2019 (COVID-19) represents a pneumonia with an unknown etiology that first appeared in Wuhan city, in the Hubei Province of China, on 31st December 2019 [1–4]. On 7th January 2020, results from Chinese research highlighted the discovery of a novel coronavirus (CoV) with a distinct genetic sequence to which they gave the name "2019-nCoV" and, subsequently, the World Health Organization (WHO) named the disease caused by this virus "COVID-19" [5,6]. After the initial outbreak in China and due to the human-to-human transmission, COVID-19 virus continued to spread worldwide, prompting the WHO first to declare a state of a global health emergency (30 January) and then to define COVID-19 as a pandemic (11 March) [7–9].

Since no vaccine has been created so far, governments have adopted strategies to limit the virus's spread [10,11]. Italy, where the first local COVID-19 case of contagion occurred on 21st February, was the first European nation to apply contrast and containment measures in order to control COVID-19's spread by declaring a state of quarantine. This led to a change in the habits of the Italian population, which included a modification to the practice of physical activity (PA). In particular, the Italian government promoted social distancing, the closure of schools and universities and the suspension of any social event, including professional and non-professional sports competitions [12]. With the first national Decree of the President of the Council of Ministers (DPCM), and the subsequent ones issued, the practice of PA for both athletes and amateurs was initially allowed by maintaining an interpersonal distance of at least one meter [12]. Then, the suspension of all activities practiced in gyms, sports centers and swimming pools and their closure was ordered [13]. Finally, to further counter the spread of the virus, access to public parks and gardens was prohibited and the practice of outdoor recreational PA was allowed near homes but not in group-based settings, and while respecting the distance of at least one meter from other people [14]. Since the containment measures implemented by the Italian government have become increasingly restrictive, the practice of PA has been progressively subjected to limiting conditions.

It is well known that physical inactivity causes over 5 million deaths worldwide and represents damage to the economy of the public health systems, which, for these reasons encourage PA for health promotion and disease prevention [15,16]. However, due to the government limitations, we hypothesized that the people would practice lower levels of PA. A standardized instrument used to assess the levels of PA practice in a population, during the "last 7 days" or in the "usual week", is the International Physical Activity Questionnaire (IPAQ) [17]. The IPAQ was developed in two different versions: the long-form (IPAQ-LF) and the short-form (IPAQ-SF), and it allows us to measure the following four intensity levels of PA: (a) vigorous; (b) moderate; (c) walking; (d) sitting [17,18]. Through the IPAQ data relating to the time spent for each PA intensity, it is possible to compute the levels of PA practice and the related weekly energy expenditure using the respective metabolic equivalent task (MET) of each PA type [17,19–21].

For this reason, the aim of our study was to measure, through an online adapted version of the IPAQ-SF, the levels of PA expressed as energy expenditure (MET–minutes/week) among the physically active Sicilian population before and during the last seven days of COVID-19 quarantine. Moreover, since previous research investigated the differences in PA practice across gender, age and body composition [22–25], in this first study we also considered the relationship between the total weekly energy expenditure before and during COVID-19 quarantine and these demographic and anthropometric variables.

## 2. Materials and Methods

### 2.1. Study Design

The present study is a quick, large cross-sectional online survey conducted using the Google Forms web survey platform (Google LLC, Mountain View, CA, USA).

### 2.2. Procedure

The online survey was anonymous and not attributable to the identity of the participants. The announcement, which included the link to the online survey, was published both on the website of the University of Palermo and of the Regional Sports School of the Italian National Olympic Committee (CONI) of Sicily. Moreover, using the snowball sampling recruitment method, the online survey was disseminated via social media such as Facebook, Instagram and WhatsApp, and shared with the personal contacts of the research group members and among the university students.

Prior to the start of the questionnaire, the online survey form comprised a brief description of the study, its purpose and the declarations of anonymity and confidentiality.

The Ethics Committees of the University of Palermo and of the Research Center on Motor Activities (CRAM) of the University of Catania (Protocol Number: CRAM-011-2020-16/03/2020) approved the study in conformity with the Declaration of Helsinki principles.

### 2.3. Participants

Participants included in the study completed the online questionnaire between the 30th of March and the 2nd of April 2020. Participants were recruited during the COVID-19 quarantine in Italy, a period in which the measures taken by the government have limited the access to PA practices in all gyms, sports centers and swimming pools, and have prohibited any outdoor PA in public parks and gardens [12–14].

Although participants that we considered for the study belong only to the Sicilian region, this type of investigation allowed us to extend the distribution of the survey across the Italian nation. A total of 1896 Italian subjects, both physically active and inactive, completed the online questionnaire. Among them, 1003 Sicilian subjects were recruited in this study.

To minimize the impact of errors due to this type of data, we adopted a cleaning process which consisted of the following steps: removal of ineligible cases and of multiple submissions of the same respondent; identification and handling of meaningless data. The latter were represented by invalid responses to the questionnaire due to the respondents' reluctance to provide valid responses and the lack of internal consistency of responses. To manage this last point, a threshold/cutoff value was calculated according to the IPAQ scoring protocol, reported in the "Guidelines for Data Processing and Analysis of the International Physical Activity Questionnaire (IPAQ)—Short and Long Forms", under the constraint of consistency of responses (http://www.ipaq.ki.se).

Hence, a number of 802 participants lived in Sicily, who declared to practice PA regularly, was included in the study.

### 2.4. Questionnaire

The online questionnaire administered to the participants was an adapted version of the IPAQ-SF that allowed us to measure the levels of PA, expressed as energy expenditure (MET–minutes/week), among the physically active Sicilian population before and during the last seven days of COVID-19 quarantine [17]. Since the questionnaire was administered to participants only once, the levels of PA for both conditions (before and during COVID-19) were assessed at the same time. The online self-reporting questionnaire consisted of 31 questions investigating the respondents' PA practice in terms of frequencies and durations of sitting, walking, moderate-intensity physical activities and vigorous-intensity physical activities. The questionnaire, reported in Appendix A, was divided into nine sections which included: (1) demographic data (questions 2 and 3); (2) anthropometric data (questions 4 and 5); (3) PA before the COVID-19 quarantine (questions 6 and 7); (4) information relating to employment and residence during COVID-19 quarantine (from question 8 to 13); (5) information (before and during the COVID-19 quarantine) relating to vigorous-intensity PA (from question 14 to 17); (6) moderate-intensity PA (from question 18 to 21); (7) walking activity (from question 22 to 25); (8) sedentary behaviors (questions 26 and 27); (9) additional information regarding the practice of PA

during the COVID-19 quarantine (from question 28 to 31). Participants included in the study were physically active, as self-reported by the question number 6.

In this first study, for statistical analysis we considered questions from section 3 and from sections 5 to 7 in order to compute the total weekly PA level before and during the COVID-19 quarantine. Moreover, questions of the sections 1 and 2 were taken into consideration in order to analyze the total weekly PA level for both conditions (before and during COVID-19) in relation to the demographic and anthropometric variables of these questionnaire parts.

*2.5. Scoring Protocol*

Based on the concept of metabolic equivalent (MET), which is equivalent to the resting metabolic rate and corresponding to 3.5 mL $O_2$ $kg^{-1}$ $min^{-1}$ or 1 kcal $kg^{-1}$ $h^{-1}$, we computed the weekly PA level, expressed as energy expenditure in MET–minutes/week (MET–min/wk) [26]. In particular, by using the basal level of energy expenditure (expressed in MET) assigned to each type of PA (the corresponding metabolic equivalent task is: 3.3 for walking; 4.0 for moderate-intensity physical activities; 8.0 for vigorous-intensity physical activities, respectively), we estimated the total weekly energy expenditure (i.e., the sum of walking, moderate-intensity physical activities and vigorous-intensity physical activities) in MET–min/wk [19–21]. The formula is the multiplication between the MET level per minutes of practice and the PA type during the last seven days. The computation of the total weekly energy expenditure using the corresponding metabolic equivalent task for each type of PA was calculated using the Compendium of Physical Activities (and subsequent updates) [19–21], and following the "Guidelines for Data Processing and Analysis of the International Physical Activity Questionnaire (IPAQ)—Short and Long Forms" (http://www.ipaq.ki.se).

As suggested by the IPAQ recommendations for scoring protocol, participants were classified into the 3 following categories of PA based on the MET–min/wk of the total weekly energy expenditure (i.e., the sum of walking, moderate-intensity physical activities and vigorous-intensity physical activities): (1) low active (<600 MET–minutes/week); (2) moderate active (≥600 MET–minutes/week); (3) high active (≥3000 MET–minutes/week) (http://www.ipaq.ki.se).

*2.6. Statistical Analysis*

The inspection of univariate distributions was conducted using descriptive statistics. To analyze survey data, some variables were re-coded. Body Mass Index (BMI) levels were classified into the following 3 categories: underweight (BMI < 18.5); normal weight (18.5 < BMI < 25); overweight (BMI > 25) [27,28]. Age classifications were grouped into 5 categories: young (≤ 25 years); young adults (25 < years ≤ 35); adults (35 < years ≤ 55); senior adults (55 < years ≤ 65); elderly (>65 years) [29–31]. Then, percentages were calculated to describe the categorical variables. In order to represent the PA level (expressed in MET–min/wk) for the categorical variables, summary statistics (percentiles, means and standard deviations) were used.

For data analysis, the following labels were assigned to the MET–min/wk variables: (a) "MET pre-COVID 19" and "MET during COVID 19" to represent the MET–min/wk before and during the COVID-19 quarantine, respectively; (b) "MET difference pre-during COVID 19" to indicate the MET–min/wk difference between before and during the COVID-19 quarantine. The "MET difference pre-during COVID 19" variable indicated the magnitude and the direction of any MET–min/wk change due to the COVID-19 quarantine. Histograms and boxplots were used to represent data analysis of these quantitative variables.

The results of the descriptive analysis have been further investigated using adequate tests in order to compare the MET–min/wk variables in relation to the demographic and anthropometric variables considered. In particular, to compare the distribution of the total weekly energy expenditure (MET–min/wk) before COVID-19 quarantine and during COVID-19 quarantine, the Wilcoxon signed-rank test for dependent groups was chosen. To analyze the relationship between gender, BMI levels, age classifications variables and the MET–min/wk variables, we carried out a bivariate

analysis. In particular, we calculated the Wilcoxon rank-sum test for the gender variable and the Kruskal-Wallis rank-sum test for BMI levels variable and for the age classifications variables. Furthermore, using the Wilcoxon rank-sum test, pairwise comparisons were calculated to analyze any significance difference between groups for BMI and age classifications. Analyses were conducted using R ver. 3.5.2 (R Core Team; Vienna, Austria) [32].

## 3. Results

### 3.1. Descriptive Analysis

#### 3.1.1. Participants

The 802 participants of the sample comprised 411 females (51%) and 391 males (49%) with the following demographic and anthropometric characteristics: mean age: $32.27 \pm 12.81$ years; height: $168.55 \pm 10.15$ cm; weight: $67.13 \pm 13.41$ kg; BMI: $23.44 \pm 3.33$ kg/m$^2$. Analysis of BMI levels allowed us to categorize participants as: underweight n = 38 (5%); normal weight n = 565 (70%); overweight n = 199 (25%). Based on age classifications used, participants of the study were grouped into: young: n = 281 (35%); young adults: n = 253 (32%); adults: n = 209 (26%); senior adults: n = 47 (6%); elderly: n = 12 (1%). All the characteristics of the participants are reported in Table 1.

**Table 1.** Characteristics of the participants.

| | Sample | |
| --- | --- | --- |
| | **n** | **%** |
| Participants | 802 | |
| Females | 411 | 51 |
| Males | 391 | 49 |
| | **Age classifications** | |
| | **n** | **%** |
| Young | 281 | 35 |
| Young adults | 253 | 32 |
| Adults | 209 | 26 |
| Senior adults | 47 | 6 |
| Elderly | 12 | 1 |
| | **BMI levels** | |
| | **n** | **%** |
| Underweight | 38 | 5 |
| Normal weight | 565 | 70 |
| Overweight | 199 | 25 |

Note: n—number; %—percentage, BMI—Body mass index.

#### 3.1.2. Energy Expenditure

Figure 1a,b show the total weekly energy expenditure in MET–min/wk of all the sample before and during the COVID-19 quarantine, respectively. Figure 1c shows the difference between MET–min/wk before and during COVID-19 quarantine conditions for all of the sample.

The related descriptive analysis carried out, reported in Table 2, showed the prevalent PA level of the participants in the before COVID-19 quarantine condition compared to during the COVID-19 quarantine (median: 3006 vs. 1483.8 MET–min/wk, respectively).

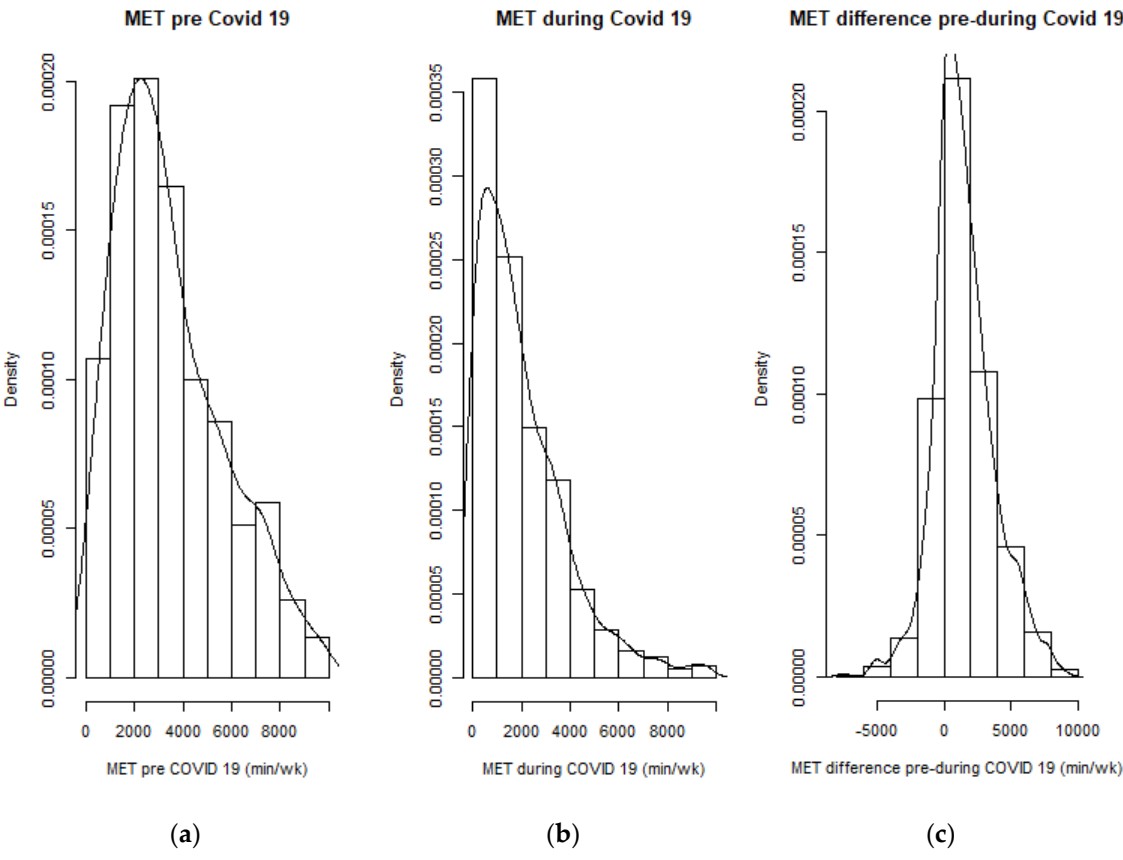

**Figure 1.** (**a**)Total weekly energy expenditure (MET-minute/week) before COVID-19 quarantine. (**b**) Total weekly energy expenditure (MET-minute/week) during COVID-19 quarantine. (**c**) Total weekly energy expenditure (MET-minute/week) difference between before and during COVID-19 quarantine.

**Table 2.** Total weekly energy expenditure in MET–minutes/week.

| Variable | Min | 1st Q | Median | Mean | 3rd Q | Max |
|---|---|---|---|---|---|---|
| MET–min/wk Before the COVID-19 quarantine | 12 | 1752 | 3006 | 3458 | 4815 | 9990 |
| MET–min/wk during the COVID-19 quarantine | 0 | 627.7 | 1483.8 | 1994.3 | 2896.1 | 9639 |
| MET–min/wk difference before and during the COVID-19 quarantine | −7440 | 61.75 | 1168.5 | 1463.51 | 2650.5 | 8934 |

Note: Min, Minimum; 1st Q, 1st Quartile; 3rd Q, 3rd Quartile; Max, Maximum; MET–min/wk, MET–minutes/week.

Table 2 shows a decrease of 1168.5 MET–min/wk from before the COVID-19 quarantine to during the COVID-19 quarantine. Using the Wilcoxon signed-rank test, we compared PA levels for these conditions and found a significant difference ($p < 0.001$).

Concerning the 3 PA categories suggested by the IPAQ recommendations for scoring protocol (i.e., $< 600$; $\geq 600$; $\geq 3000$ MET–min/wk), responses analysis for the before COVID-19 quarantine conditions showed 49 low active participants (6%); 352 moderately active participants (44%); and 401 high active participants (50%); meanwhile, the during COVID-19 quarantine condition results showed an increase of 19% of low active participants (n = 200) and an increase rate of 7% of moderately active participants (n = 409), with a related decrease of 26% of high active participants (n = 193).

A comparison between MET–min/wk before and during the COVID-19 quarantine showed a decrease of the total weekly energy expenditure during the COVID-19 quarantine for the 77% of the sample (n = 615).

### 3.2. Energy Expenditure in Relation to Gender, BMI Levels, and Age Classifications: A Bivariate Analysis

Figures 2–4 show the MET–min/wk comparison before and during the COVID-19 quarantine in relation to the gender, BMI and age groups, respectively. Boxplots show the differences between the distributions of the variables considered, which were analyzed separately, as reported below.

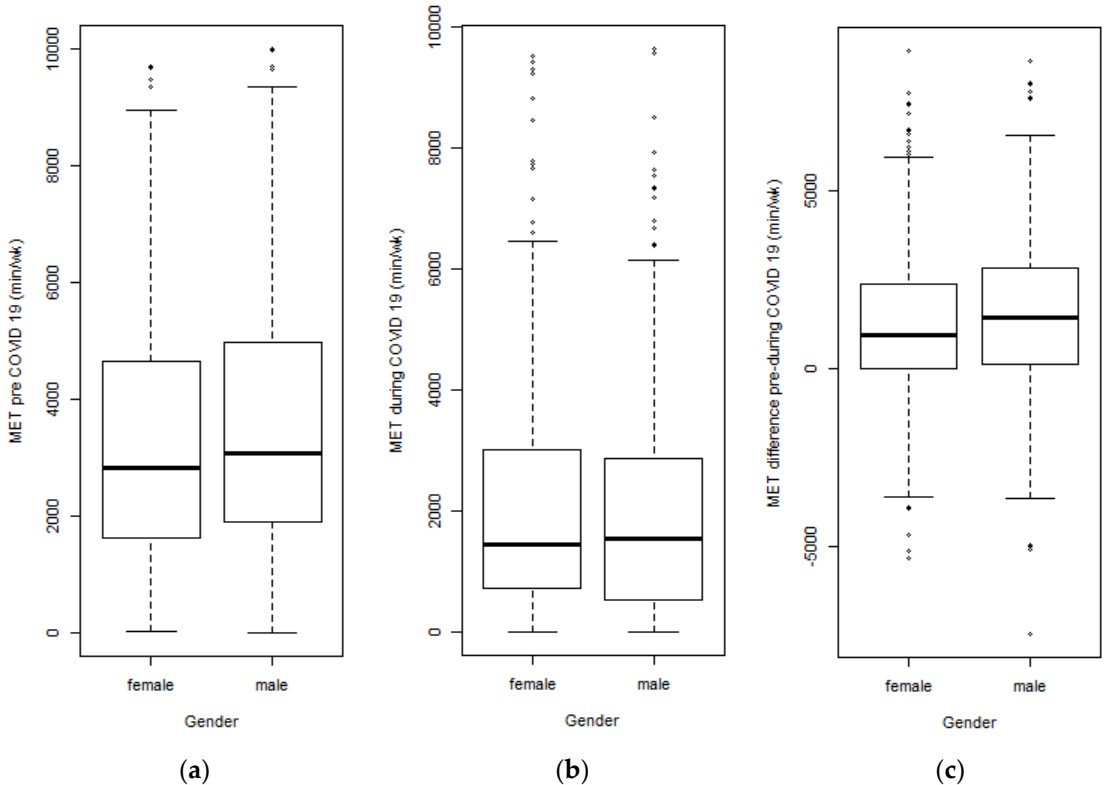

**Figure 2.** (**a**) Box plots (median, interquartile range) describe the total weekly energy expenditure (MET-minutes/week) before COVID-19 quarantine in relation to the gender variable. (**b**) Box plots (median, interquartile range) describe the total weekly energy expenditure (MET-minutes/week) during COVID-19 quarantine in relation to the gender variable. (**c**) Box plots (median, interquartile range) describe the total weekly energy expenditure (MET-minutes/week) difference between before and during COVID-19 quarantine in relation to the gender variable.

For gender, male participants showed a distribution that shifted to higher values of MET–min/wk compared to female participants before the COVID-19 quarantine (Figure 2a). During the COVID-19 quarantine, although both groups reduced their total weekly energy expenditure, male and female groups showed an opposite trend than the previous one: the males distribution was slightly shifted to lower values compared to females (Figure 2b). The absolute variation of MET–min/wk (i.e., the MET–min/wk difference between before and during the COVID-19 quarantine) showed the greatest absolute decrease for males, though the 25% of participants had increased the level of PA, regardless of gender (Figure 2c).

Regarding the BMI levels, analysis showed that before the COVID-19 quarantine, the normal weight group reported the highest median value, while the underweight and overweight groups showed superimposable values (Figure 3a). We also observed that before the COVID-19 quarantine, the greatest variability of PA level distribution appeared for overweight participants, in terms of interquartile difference and range. During the COVID-19 quarantine, all groups showed a reduction

in MET–min/wk, while underweight and normal weight participants maintained the shape of the distributions observed before the COVID-19 quarantine, while the distribution of the overweight participants underwent a significant change, with 75% of cases placed on metric levels below 2000 MET–min/wk. That is, the overweight group showed the lowest median PA level during COVID-19 quarantine compared to the other groups (Figure 3b). For the distribution of the MET–min/wk absolute variation for BMI levels, it was revealed that the values of the first and third quartiles were slightly higher for overweight participants compared to those of the other groups, showing a greater contraction of the PA level. The highest MET–min/wk difference between before and during the COVID-19 quarantine was found for the overweight group (Figure 3c).

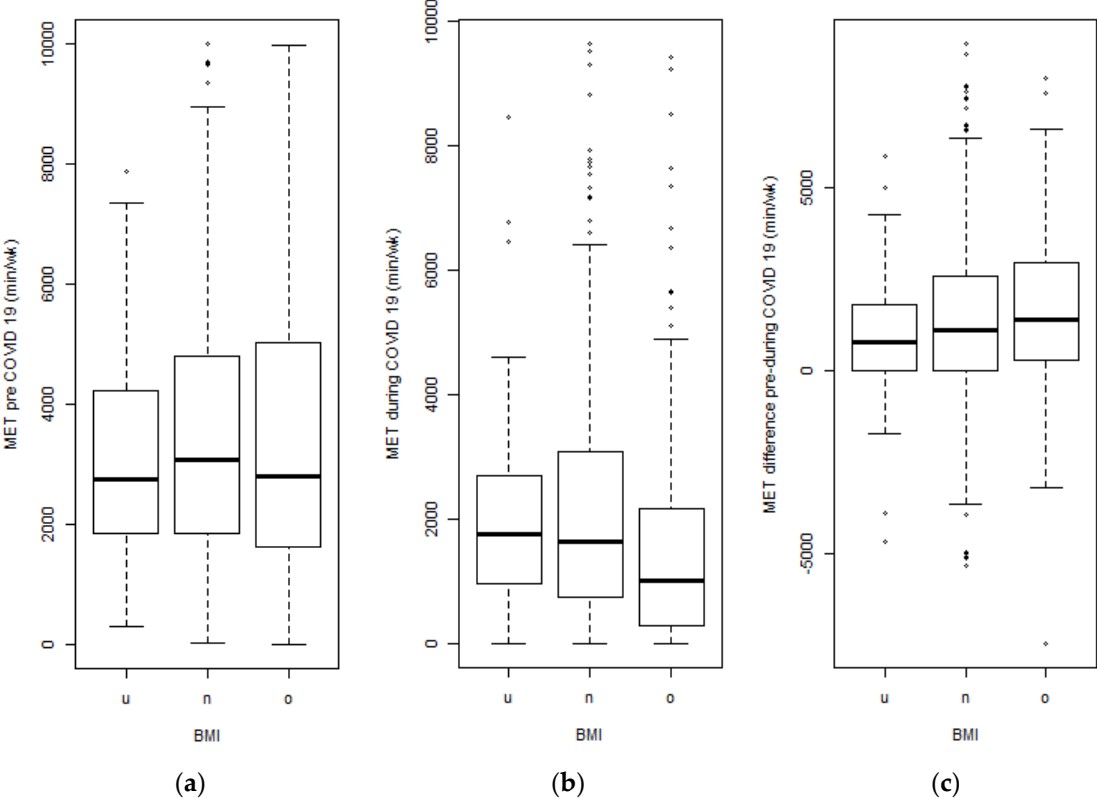

|  (a)  |  (b)  |  (c)  |

**Figure 3.** (**a**) Box plots (median, interquartile range) describe the total weekly energy expenditure (MET-minutes/week) before COVID-19 quarantine in relation to the BMI variable. Legend: u, underweight; n, normal weight; o, overweight. (**b**) Box plots (median, interquartile range) describe the total weekly energy expenditure (MET-minutes/week) during COVID-19 quarantine in relation to the BMI variable. Legend: u, underweight; n, normal weight; o, overweight. (**c**) Box plots (median, interquartile range) describe the total weekly energy expenditure (MET-minutes/week) difference between before and during COVID-19 quarantine in relation to the BMI variable. Legend: u, underweight; n, normal weight; o, overweight.

Regarding the age classifications, the comparison between before and during the COVID-19 quarantine showed a MET–min/wk distribution characterized by different location and dispersion parameters for each group. Figure 4a,b show an inverse relationship between PA level and age. Moreover, as observed by the same figures, all groups showed a reduced the MET–min/wk in a similar way. During the COVID-19 quarantine, the median of the elderly group was about 2/3 of the median of young participants and about 1/2 of the median of adults. As shown in Figure 4c, the differences in the location parameters referring to the individual groups were significantly reduced. Participants belonging to the young, young adults and adults groups showed higher MET–min/wk differences between before and during the COVID-19 quarantine, as shown in Figure 4c.

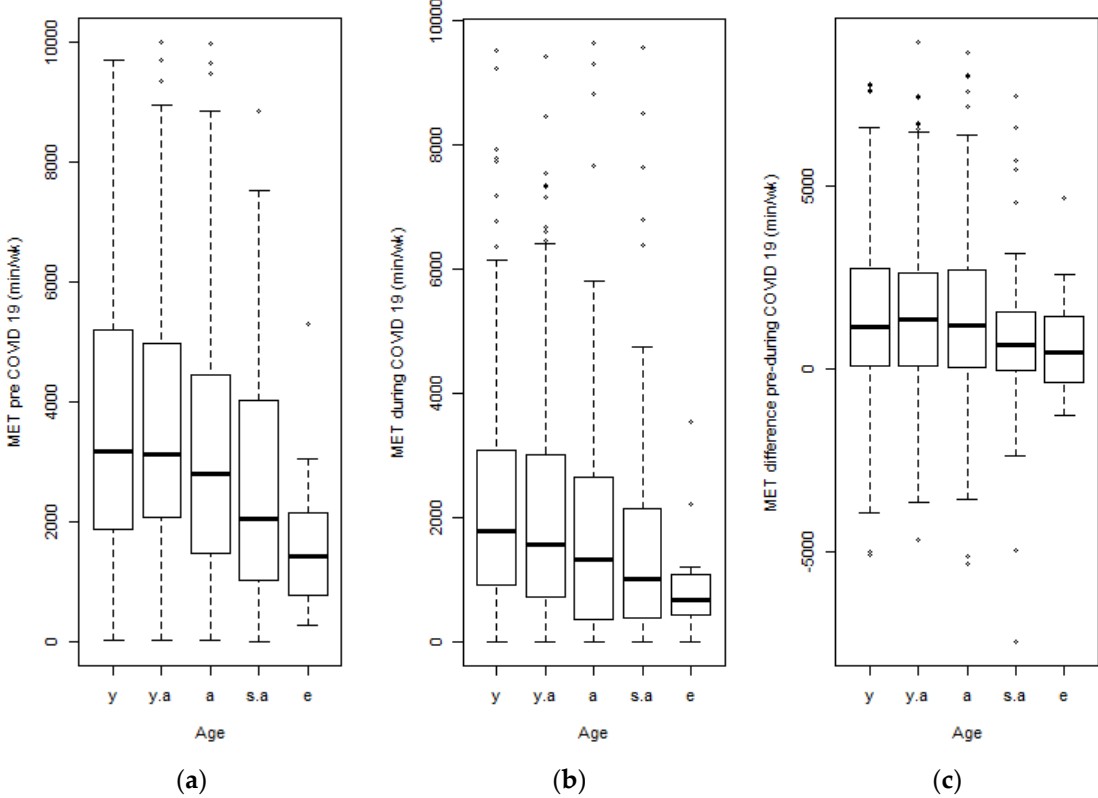

**Figure 4.** (**a**) Box plots (median, interquartile range) describe the total weekly energy expenditure (MET-minutes/week) before COVID-19 quarantine in relation to the age variable. Legend: y, young; y.a, young adults; a, adults; s.a, senior adults; e, elderly. (**b**) Box plots (median, interquartile range) describe the total weekly energy expenditure (MET-minutes/week) during COVID-19 quarantine in relation to the age variable. Legend: y, young; y.a, young adults; a, adults; s.a, senior adults; e, elderly. (**c**) Box plots (median, interquartile range) describe the total weekly energy expenditure (MET-minutes/week) difference between before and during COVID-19 quarantine in relation to the age variable. Legend: y, young; y.a, young adults; a, adults; s.a, senior adults; e, elderly.

The Wilcoxon rank-sum test was calculated to evaluate the null hypothesis that male and female distributions of the total weekly energy expenditure were similar. The bivariate analysis between gender and the MET–min/wk variables showed a significant difference before the COVID-19 quarantine ($p = 0.046$) and in the MET difference between before and during the COVID-19 quarantine ($p = 0.009$) for male and female. Regarding the BMI findings, we found a significant difference during COVID-19 ($p < 0.001$). Pairwise comparisons showed a significant difference between underweight and overweight groups ($p = 0.025$) and between normal weight and overweight groups ($p < 0.001$) during the COVID-19 quarantine. Age classifications analysis that was carried out showed a significant difference between before and during the COVID-19 quarantine ($p < 0.001$ for both conditions). Pairwise comparisons showed a significant difference between young and adults ($p = 0.003$), young and elderly ($p = 0.012$) and young adults and elderly ($p = 0.047$) in MET–min/wk before the COVID-19 quarantine, as well as during COVID-19 quarantine between young and adults ($p = 0.022$) and between young and elderly ($p = 0.05$).

## 4. Discussion

The aim of the present study was to assess the level of PA through an adapted version of the IPAQ-SF, expressed as energy expenditure (MET–min/wk) among the physically active Sicilian population before and during the last seven days of the COVID-19 quarantine. Notably, due to the quarantine and containment measures adopted by the Italian government in order to control the spread of COVID-19, the practice of PA was subject to significant restrictions [12–14]. As a result,

we hypothesized that these limitations have induced the population to decrease their habitual PA level during the quarantine.

The hypothesis we formulated was confirmed because our findings showed significantly lower levels of PA among the physically active Sicilian population during the COVID-19 quarantine compared to before the COVID-19 quarantine.

We suppose that among the possible causes of the lower levels of PA that we found could be the following: absence of coach / personal trainer / instructor / training partner; lack of equipment; insufficiency of large spaces usually available for PA; different setting. Indeed, a number of studies investigated the importance of different exercise features such as the exercise type, exercise mode, exercise setting and level and quality of supervision [33–37]. However, during the current quarantine, the population was induced to modify their practices of PA in a home-based setting and, although interest in home-based exercise is growing in parallel with COVID-19's spread around the world, the population could be unable to adapt their regular training to their homes [38].

The abrupt reduction or interruption of PA induces several effects in the human body both in the short- and long-term [39,40]. Firstly, the drastic decrement of PA leads to acute modifications such as atrophy and muscle mass reduction in a few days [41,42]. Prolonged periods of lower levels of PA can cause a failure to maintain body weight, increasing health risks [43]. Lastly, a long-term sedentary lifestyle induces to adaptations that negatively affect the cardiorespiratory fitness and metabolic profile, features related to the prevention of several diseases [44,45].

Due to the benefits of PA on psycho-physiological human functions that in this critical period could be compromised, major institutions as the World Health Organization (WHO) and the American College of Sports Medicine (ACSM) developed guidelines for healthy people, recommending specific exercise programs and daily strategies to adopt during the quarantine [46,47]. Furthermore, many research groups have developed different home-based training programs, suggesting a selection of exercises in order to maintain a physically active lifestyle during the current emergency [38,48].

The relation between the MET–min/wk before and during the COVID-19 quarantine and the demographic and anthropometric variables that we considered in this study showed interesting findings.

Regarding the gender variable, we found a significant difference in the pre-COVID-19 quarantine condition and in the MET difference between before and during the COVID-19 quarantine. The male group showed a significantly greater total weekly energy expenditure before the COVID-19 quarantine, and a significantly greater MET–min/wk difference between before and during the COVID-19 quarantine than the female group. The differences in PA between gender that we found are supported by the scientific literature [22]. In fact, the greater PA level before the COVID-19 quarantine for male group indicated, firstly, that males habitually train with a higher level of frequency and volume for any intensity of PA [22]. Furthermore, since the COVID-19 quarantine began, the PA levels were superimposable for both gender groups; in agreement with Li et al., we suppose that males prefer outdoor activities and they train less often in home-based settings compared to females [49]. Moreover, the 25% of participants increased their level of PA during the COVID-19 quarantine, regardless of their gender. The latter finding could be due to the availability of more free time during the quarantine period, in line with the literature reporting that the lack of free time represents an obstacle to the practice of PA [50,51].

As for the BMI variable, we found a significant difference during the COVID-19 quarantine. However, it is necessary to recall that the sample, as reported by the respondents, was physically active. In fact, for the recruitment of the sample, we used the website of the Regional Sports School of the Italian National Olympic Committee (CONI) of Sicily and the snowball sampling method among the students of the Sport Sciences Faculty of the University of Palermo. It is widely recognized that BMI is a parameter that does not allow us to distinguish a prevalence of muscle mass or fat mass in overweight conditions [52]. In particular, the overweight level in some athletes is due to a greater muscle mass for the type of sport practiced (e.g., weightlifting, powerlifting, bodybuilding) [52]. This issue is taken into consideration for the assessment of the body composition of athletes during which, in order

to determine the percentage of muscle mass and fat mass, other evaluations such as the Bioelectric Impedance Analysis (BIA) are commonly used [52]. Based on these premises, we suppose that the overweight group of our sample, being physically active, was composed of subjects with a greater muscle mass. This assumption could be supported by the fact that these subjects showed a high level of total weekly energy expenditure before the COVID-19 quarantine. Moreover, they were the ones who experienced the highest decrease during the COVID-19 quarantine and therefore the highest MET–min/wk difference between before and during the COVID-19 quarantine. The latter outcome can be explained as being due to a lack of equipment.

For the age classifications variable, we found a significant difference before and during the COVID-19 quarantine in which the highest and the lowest level of PA, both before and during the COVID-19 quarantine, were reported by the young and elderly, respectively. These findings are in agreement with the work by Li et al., in which they stated that the practice of PA decreases progressively with aging, as shown by the trend of our results [49]. Similarly, Hallal et al. reported that younger subjects are more physically active than older ones [53]. Moreover, our outcomes showed that the young, young adults and adults groups reported higher MET–min/wk differences between before and during the COVID-19 quarantine compared to the other age groups. Since physical inactivity is positively related to a higher rate of morbidity and an increase of incidence of mortality, especially for some sections of the population such as the elderly [54,55], the scientific literature reinforced the importance of maintaining an active physical state, even during the current pandemic [56–59]. In fact, it is widely recognized that PA induces positive effects on different psycho-physiological aspects at every stage of life [60–62]. For instance, among the benefits of PA, it has been demonstrated that it is important for the peak bone mass in adolescents [63], and that it plays a key role in reducing the risk of cardiovascular disease in adults [64], in preventing the risk of falling [65] and in counteracting frailty and sarcopenia in the elderly [66]. Among the different age groups, the elderly represent the population most susceptible to a significant reduction in physical activity due to the physiological ageing-related decline of the musculoskeletal system to which they are subject [54].

### 4.1. Future Research Goals

For this first study, we limited ourselves to analyzing the total weekly energy expenditure before and during the COVID-19 quarantine and the relation of the latter with specific demographic and anthropometric variables. The future goal of the research, which will continue until the end of the quarantine, will be to evaluate any differences in the weekly energy expenditure for each type of PA, which are: (a) vigorous; (b) moderate; (c) walking; (d) moderate to vigorous, as well as in relation to all the other variables of the presently adapted IPAQ-SF.

Moreover, our findings seem to suggest that a home-based setting is not favorable to the practice of PA. Since one of our suppositions is that the insufficiency of large spaces for PA could be one of the most influential factors in the reduction of PA, our further study will aim to extend the analysis using multivariate models to investigate the MET–min/wk variability with respect to the other variables of the questionnaire that analyzed the conditions in which PA is carried out (e.g., house extension, availability of outdoor spaces, etc.).

### 4.2. Future Practical Applications

A key issue concerning the adaptation of PA during the quarantine could be its future practical application related to different emergency conditions (e.g., public health emergencies, lockdown for natural environmental conditions, adverse weather conditions). In fact, some sports facilities are experimenting and adopting, for the type of activities that allow it, with remote fitness lessons [67,68]. If this experimental approach will lead to achievement of the set goals, then some fitness centers may adopt it because it is a sustainable form of PA and provides easy access for the entire population.

### 4.3. Strengths and Limitations of the Study

The key strengths of the research are the typology of the study we appropriately selected during this period and its ease of access for the participants. In fact, due to the quarantine, the online survey is an ideal research instrument, as it allowed us to recruit a large sample. Among the strengths of the online survey, we highlight the possibility of reaching the population belonging to different geographical areas and, moreover, the speed of data collection.

Among the main limitations of our study, it is necessary to account for the possibility of the over-reporting bias of PA, which is common among the respondents of a self-reported questionnaire [69,70]. However, since our questionnaire asked respondents to indicate the level of PA for two different periods (i.e., before and during the COVID-19 quarantine), we speculate that the internal consistency of the respondents led them to report the same bias both for the questions covering before and during the COVID-19 quarantine. For this reason, the MET–min/wk difference between before and during COVID-19 quarantine was not affected. The limitations of the study also include the self-selection bias that could cause a nonprobability sampling, affecting the generalisability of the results. Nevertheless, our outcomes are in agreement with the literature.

## 5. Conclusions

Based on our outcomes, we can determine that the current quarantine has negatively influenced, and has greater impacts on the practice of PA, especially for males and overweight subjects. Moreover, regarding age classifications, the young, young adults and adults groups were more affected than others.

Although we sustain the idea that doing at least some PA is better than doing nothing [71], we promote the concept that practising more PA is better than practising less of it [71]. For this reason, we believe it is possible to increase the level of home-based training during quarantine. We recommend following the scientific guidelines, asking to a personal trainer for personalized training and consulting a doctor in sports sciences in order to maintain a habitual PA level.

Furthermore, since the training load plays a key role in musculoskeletal injuries, we suggest to maintaining PA level closer to the habitual level and gradually returning to the previous training load [72]. Moreover, in critical situations such as the current pandemic, encouraging the practice of PA for both physically active and inactive populations by sports professionals (coaches /personal trainers/instructors/graduate doctors in sport and exercise sciences), as well as by medical doctors, should be the main way to make the population aware of the need for maintaining an active state for health promotion.

**Author Contributions:** Conceptualization and Methodology, G.B.; Questionnaire Development, G.B.; Software (uploading the questionnaire to the Google Forms online platform and dissemination of the survey link), A.G.; Software (dissemination of the survey link), G.M.; Data analysis, A.M.P.; Data Interpretation, A.M.P., V.G., G.B.; Writing—Original Draft Preparation, V.G.; Writing—Review & Editing, A.M.P., G.B.; Project Administration and Supervision, A.P., G.B. All authors have read and agreed to the published version of the manuscript.

**Funding:** This research received no external funding.

**Conflicts of Interest:** The authors declare no conflict of interest.

## Appendix A

**Table A1.** Online survey on physical activity levels before and during the last 7 days of the coronavirus disease 2019 (COVID-19) quarantine.

**Demographic and Anthropmetric Data**

| N. | Question | Options |
|----|----------|---------|
| 1 * | Is the first time you completed this survey? | Yes; No |
| 2 * | What is your age? (years) | 0–10; 11–15; 16–20; 21–25; 26–30; 31–35; 36–40; 41–45; 46–50; 51–55; 56–60; 61–65; 66–70; 71–75; 76–80; 81–85; 86–90; 91–95; 96–100; >100; I don't want to answer |
| 3 * | Are you male or female? | Male; Female |
| 4 * | What is your weight? (kg) | 10–20; 20–30; 30–40; 41–45; 46–50; 51–55; 56–60; 61–65; 66–70; 70–75; 76–80; 81–85; 86–90; 91–95; 96–100; 101–105; 106–110; >110; I don't want to answer |
| 5 * | What is your height? (cm) | <100; 100–110; 111–120; 121–130; 131–135; 136–140; 141–145; 146–150; 151–155; 156–160; 161–165; 166–170; 171–175; 176–180; 181–185; 186–190; 191–195; 196–200; >200; I don't want to answer |
| 6 * | Before the COVID-19 quarantine, did you train regularly? | Yes; No |
| 7 * | Before the COVID-19 quarantine, how many days/week did you train regularly? | 0; 1; 2; 3; 4; 5; 6; 7 |
| 8 * | In which Italian region or foreign country do you currently live? | Valle d'Aosta; Piemonte; Liguria; Lombardia; Trentino-Alto Adige; Veneto; Friuli-Venezia Giulia; Emilia-Romagna; Toscana; Umbria; Marche; Marche; Lazio; Abruzzo; Molise; Campania; Puglia,; Basilicata; Calabria; Sicilia; Sardegna; Other |
| 9 * | Please, indicate the type of work done during this period: | Work in remote /Work at home; On my workplace; Government measures do not allow me to work; I am retired /I am sick /I was laid-off; I am a student; Other |
| 10 * | Where do you currently live? | City; Suburb; Other |
| 11 * | Please, indicate the house type you live in during this period: | Apartment/Condominium; Multi-storey detached house; Villa/Chalet/ Multi-family villa house/Terraced house; Other |
| 12 * | Please, indicate the size of the property you live in during this period (m$^2$): | 0–50; 50–100; 100–200; 200–400; 400–800; Other |
| 13 * | Does your property have outdoor spaces for physical activity? | Yes; No |

**Table A1.** *Cont.*

| N. | Question | Options | | | | | | | |
|---|---|---|---|---|---|---|---|---|---|
| | | **Demographic and Anthropmetric Data** | | | | | | | |
| | | **Vigorous-Intensity Physical Activity**<br>Vigorous-intensity physical activities are those that require strenuous physical effort<br>and that increase the breathing rate more frequently than normal.<br>Please think only about those physical activities that you performed for at least 10 consecutive minutes. | | | | | | | |
| 14 * | Before the COVID-19 quarantine,<br>how many days/week did you perform vigorous-intensity physical activities, such as lifting heavy objects, hoeing the earth,<br>practicing zumba, cycling on an exercise bike, or running on a treadmill at high speed?<br>Please indicate the number of days per week: | 1 | 2 | 3 | 4 | 5 | 6 | 7 | No vigorous-intensity physical activity |
| 15 * | Before the COVID-19 quarantine,<br>how long in total did you usually spend in vigorous-intensity physical activities on a day of the week?<br>Please indicate the total number of minutes: | | | | _______ (Total minutes) | | | | |
| 16 * | During the COVID-19 quarantine,<br>how many days, in the last 7 days, did you perform vigorous-intensity physical activities, such as lifting heavy objects,<br>hoeing the earth, practicing zumba, cycling on an exercise bike or running on a treadmill at high speed?<br>Please indicate the number of days in the last week: | 1 | 2 | 3 | 4 | 5 | 6 | 7 | No vigorous-intensity physical activity |
| 17 * | During the COVID-19 quarantine,<br>how long in total did you spend performing vigorous-intensity physical activities on a day of the last week?<br>Please indicate the total number of minutes: | | | | _______ (Total minutes) | | | | |
| | | **Moderate-Intensity Physical Activity**<br>Moderate-intensity physical activities are those that require a modest physical effort<br>and that increase the breathing rate.<br>Please think only about those physical activities that you performed for at least 10 consecutive minutes. | | | | | | | |
| 18 * | Before the COVID-19 quarantine,<br>how many days/week did you perform moderate-intensity physical activities, such as carrying light objects, working in the<br>garden, going to the gym, cycling at a regular pace or doing prolonged physical work at home? Please do not include<br>walking.<br>Please indicate the number of days per week: | 1 | 2 | 3 | 4 | 5 | 6 | 7 | No moderate-intensity physical activity |
| 19 * | Before the COVID-19 quarantine,<br>how long in total did you usually spend in moderate-intensity physical activities on a day of the week?<br>Please indicate the total number of minutes: | | | | _______ (Total minutes) | | | | |
| 20 * | During the COVID-19 quarantine,<br>how many days, in the last 7 days, did you perform moderate-intensity physical activities, such as carrying light objects,<br>working in the garden, going to the gym, cycling at a regular pace, or doing a prolonged physical work at home? Please do<br>not include walking.<br>Please indicate the number of days in the last week: | 1 | 2 | 3 | 4 | 5 | 6 | 7 | No moderate-intensity physical activity |
| 21 * | During the COVID-19 quarantine,<br>for how long in total did you spend performing moderate-intensity physical activities on a day of the last week?<br>Please indicate the total number of minutes: | | | | _______ (Total minutes) | | | | |
| | | **Walking Activities**<br>This section evaluates the total time spent in walking activities both at work and at home to move from one place to another, and any other walking you have done just for fun, as a hobby, for shopping or for performing exercise | | | | | | | |
| 22 * | Before the COVID-19 quarantine,<br>how many days/week did you walk for at least 10 consecutive minutes?<br>Please indicate the number of days per week: | 1 | 2 | 3 | 4 | 5 | 6 | 7 | No day |
| 23 * | Before the COVID-19 quarantine,<br>how long in total did you usually spend on walking activities on a day of the week?<br>Please indicate the total number of minutes: | | | | _______ (Total minutes) | | | | |
| 24 * | During the COVID-19 quarantine,<br>during how many days, in the last 7 days, did you walk for at least 10 minutes continuously?<br>Please indicate the number of days in the last week: | 1 | 2 | 3 | 4 | 5 | 6 | 7 | No day |

**Table A1.** *Cont.*

| N. | Question | Options | | | | |
|---|---|---|---|---|---|---|
| | | **Demographic and Anthropmetric Data** | | | | |
| 25 * | During the COVID-19 quarantine, how long in total did you spend on walking activities on a day of the last week? Please indicate the total number of minutes: | __________ (Total minutes) | | | | |
| | | **Sitting Activities** This section evaluates the total time spent on sitting activities both at work and at home, and any other sitting activities you have done to attend a course or during free time. The latter include the time sitting / lying down reading or watching TV | | | | |
| 26 * | Before the COVID-19 quarantine, how long in total did you usually spend on sitting activities on a day of the week? Please indicate the total number of minutes: | __________ (Total minutes) | | | | |
| 27 * | During the COVID-19 quarantine, how long in total did you spend in sitting activities on a day of the last week? Please indicate the total number of minutes: | __________ (Total minutes) | | | | |
| | | **The COVID-19 quarantine and home physical activity** This section evaluates how you perform physical activities at home during the COVID-19 quarantine | | | | |
| 28 * | During the COVID-19 quarantine, which kind of physical activities did you perform in your home in the last 7 days? | Exercises without equipment (body weight exercises) | Exercises with equipment (balls, bands, dumbbells, barbells, etc.) | None of the above | I don't practice any physical activity | Other |
| 29 * | During the COVID-19 quarantine, in the last 7 days, did you practice physical activities while listening to music in your home? | Yes | | No | | I don't practice any physical activity |
| 30 * | During the COVID-19 quarantine, in the last 7 days, in order to plan / perform physical activities at home, you used: | Web sites | Personal experiences | Advice from a coach | Tutorial | I don't practice any physical activity | Other |
| 31 * | During the COVID-19 quarantine, in the last 7 days, how did you perform physical activities at home? | Alone | | In pair | In small group | I don't practice any physical activity |

Note: * Required field.

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
