# Peer review of "Physical Activity Levels and Related Energy Expenditure during COVID-19 Quarantine among the Sicilian Active Population: A Cross-Sectional Online Survey Study"

_sustainability, doi:10.3390/su12114356_

Round 1

Reviewer 1 Report

The authors would like to show the effects of the quarantine on daily physical activity in Italy. The approach of working with an online survey during this time is appropriate.

It is not surprising that the physical activity decreases as part of the quarantine actions. However, it has also been shown that some individuals have shown a corresponding increase in activity.

More important to highlight is the consequences of the lack of activity in the different age groups. For example, activity in the elderly decreases significantly and leads to increased immobility with the risk of frailty. In addition, women do not seem to show any major changes in activity. This may be due to a generally lower activity, the habit of doing more sports inside or the fact that the daily routine has not really changed.

The authors have to re-examine the manuscript intensively with regard to grammar and spelling.

The illustrations are difficult to read and there is no clear explanation which statistical tests were used. There is also a lack of signs of significance in the figures and table 2. The p-values are displayed differently within the text.

Author Response

Reviewer #1

The authors would like to show the effects of the quarantine on daily physical activity in Italy. The approach of working with an online survey during this time is appropriate.

Dear Reviewer, thank you very much. We appreciate.

It is not surprising that the physical activity decreases as part of the quarantine actions. However, it has also been shown that some individuals have shown a corresponding increase in activity.

We agree with the Reviewer. As we reported in the results section some participants increased the PA level.

More important to highlight is the consequences of the lack of activity in the different age groups. For example, activity in the elderly decreases significantly and leads to increased immobility with the risk of frailty. In addition, women do not seem to show any major changes in activity. This may be due to a generally lower activity, the habit of doing more sports inside or the fact that the daily routine has not really changed.

We thank the Reviewer for the positive comment and following the suggestion we have discussed the negative consequences of the lack of physical activity at every stage of the life, focusing more on elderly. Moreover, we added information about the short- and long-term effects in human body of the abrupt reduction or interruption of physical activity.

The authors have to re-examine the manuscript intensively with regard to grammar and spelling.

We thank the Reviewer for the comment that improves the quality of the manuscript. We have carefully examined the English language and the style of the entire manuscript.

The illustrations are difficult to read and there is no clear explanation which statistical tests were used. There is also a lack of signs of significance in the figures and table 2. The p-values are displayed differently within the text.

We thank the Reviewer for raising these points. In this revised version of the manuscript, we better explaining the analysis carried out. Moreover, the results section has been reviewed and the captions of the figures have been changed. Furthermore, all the p-values reported have been checked throughout the manuscript.

Reviewer 2 Report

Good study showing logical consequence of quarantine  on physical activity. Methods are properly selected. The only comment is that some information is presented twice (in tables and figures) that increase the length of the article.

Author Response

Reviewer #2

Good study showing logical consequence of quarantine on physical activity. Methods are properly selected. The only comment is that some information is presented twice (in tables and figures) that increase the length of the article.

Dear Reviewer, thank for the comment. We appreciate. We agree with the reviewer regarding the fact that Figure 1 and Table 3 reported information of the same variables. However, we would like to maintain Table 3 in the manuscript because it allows to know the position indices of the variables that from figure 1 alone it would not be possible to know.

Reviewer 3 Report

Generally, I feel this is a very interesting study. Moreover, it is very timely and focuses on a really important issue. It is fairly well-written, mostly clear and the most of the conclusions are sound. There is, however, one issue that has to be resolved before the study can be accepted for publication. The authors failed to discuss the limitations of the study. There are several large drawbacks inherent to the study design and implications of theses on the generalisability of the results needs to be discussed.

I also have several comments that need addressing:

Abstract:

state that PA was measured only once, i.e. that PA before COVID-19 was assessed at the same time as during COVID-19.

l.26: what does demo-antropometric mean? Consider rephrasing.

l-36-41: describe the direction of these differences. Which groups had the largest reduction in PA? Currently, it is not clear from the abstract how did you arrive to the conclusions you listed. In addition, I believe that your results do not support the conclusion that younger groups are affected more than the others (although p value for this is not given in the results). Be sure to amend this part of the conclusion section (both in abstract and in the main text).

Methods:

l.126: describe more clearly how did the participants report they engaged in regular PA (i.e. that this was a specific question before the IPAQ questionnaire.

l.173-176: describe the cut-offs used for BMI categories and age groups

Results:

include explanation about what the lines in the box plots stand for.

l.285-295. I suggest omitting comparisons between genders and BMI and age groups relating to PA level before or during COVID-19. Stay focused on differences in the change in PA and report only this. Give numbers on the median change in each group.

Discussion:

l.315-317: this sentence is not clear. What were you trying to say? Rephrase please.

l.318-333: this paragraph is much too long. I do not see how it fits in the discussion. It should probably be just 1-2 sentences and it should be moved to conclusions or practical applications

list strengths and limitations of the study. Consider all the biases inherent to the study design.

Author Response

Reviewer #3

Generally, I feel this is a very interesting study. Moreover, it is very timely and focuses on a really important issue. It is fairly well-written, mostly clear and the most of the conclusions are sound. There is, however, one issue that has to be resolved before the study can be accepted for publication. The authors failed to discuss the limitations of the study. There are several large drawbacks inherent to the study design and implications of theses on the generalisability of the results needs to be discussed.

Dear Reviewer, thank you. We appreciate.

We apologize for this forgetfulness. In agreement with the Reviewer’s suggestion, in the new version of the manuscript we included the paragraph regarding strengths and limitations of the study, reporting the known limitations for this type of studies.

I also have several comments that need addressing:

Abstract:

state that PA was measured only once, i.e. that PA before COVID-19 was assessed at the same time as during COVID-19.

We thank the Reviewer for raising this important suggestion. Due to limit of the words for the abstract section we have added this information in the methods section of the manuscript (paragraph 2.4). However, abstract has been partially re-written in order to make it clearer.

l.26: what does demo-antropometric mean? Consider rephrasing.

We apologize for the unclear explanation. In the revised version of the manuscript we have now substituted the word “demo-antropometric” with the words “demographic and antropometric” through the whole manuscript.

l-36-41: describe the direction of these differences. Which groups had the largest reduction in PA? Currently, it is not clear from the abstract how did you arrive to the conclusions you listed. In addition, I believe that your results do not support the conclusion that younger groups are affected more than the others (although p value for this is not given in the results). Be sure to amend this part of the conclusion section (both in abstract and in the main text).

We apologize for the unclear explanation. In order to add the information that the Reviewer suggested, we computed a pairwise comparisons between groups using the Wilcoxon rank-sum test. Then, we reported these results both in the abstract and in the results section of the manuscript. Furthermore, we reviewed the conclusion both in the abstract and in the main text.

Methods:

l.126: describe more clearly how did the participants report they engaged in regular PA (i.e. that this was a specific question before the IPAQ questionnaire).

We thank the Reviewer for highlighting this point. Yes, participants declared to be physically active or inactive answering to the question number 6 of the questionnaire. Following the reviewer’s suggestion, we added this information in the paragraph 2.4 of the main text. 

l.173-176: describe the cut-offs used for BMI categories and age groups

We thank the Reviewer for the positive comment that improves the manuscript clarity. In the new version of the manuscript, in the paragraph 2.6 we have added the cut-offs used for both the BMI and the age groups.

Results:

include explanation about what the lines in the box plots stand for.

We thank the Reviewer for this suggestion. The captions of the all boxplots have been modified in the reviewed version of the manuscript.

l.285-295. I suggest omitting comparisons between genders and BMI and age groups relating to PA level before or during COVID-19. Stay focused on differences in the change in PA and report only this. Give numbers on the median change in each group.

We thank the Reviewer for the suggestion. However, we consider it appropriate to also report comparisons between genders, BMI, and age groups relating to PA level before and during COVID-19 for a complete explanation of the statistical analysis carried out. Median change in each group was included in the revised version of the manuscript.

Discussion:

l.315-317: this sentence is not clear. What were you trying to say? Rephrase please.

We apologize for the unclear explanation. In the revised version of the manuscript we have hopefully explained more clearly all the section.

l.318-333: this paragraph is much too long. I do not see how it fits in the discussion. It should probably be just 1-2 sentences and it should be moved to conclusions or practical applications

Following this insightful suggestion of the Reviewer, the sentence has been rewritten and it has been moved to the conclusion section.

list strengths and limitations of the study. Consider all the biases inherent to the study design.

We thank the Reviewer for raising this important suggestion and we apologize for this forgetfulness. In the revised version of the manuscript we included strengths and limitations of the study (paragraph 4.3).

Round 2

Reviewer 1 Report

The authors replied to every comment. Thank you very much for your revision work.

Reviewer 3 Report

The revised manuscript is much improved. Most of my concerns were well addressed. I look forward to seeing future studies from the authors